# Immune Regulation by Cytosolic DNA Sensors in the Tumor Microenvironment

**DOI:** 10.3390/cancers15072114

**Published:** 2023-04-01

**Authors:** Keitaro Fukuda

**Affiliations:** 1RIKEN Center for Integrative Medical Sciences (IMS), Laboratory for Skin Homeostasis, Yokohama 230-0045, Japan; keitaro.fukuda@riken.jp; 2Department of Dermatology, Keio University School of Medicine, Tokyo 160-8582, Japan

**Keywords:** cytosolic DNA sensor, cGAS, STING, type I IFN, AIM2, IL-1β, IL-18, TIDC, immunotherapy, tumor microenvironment

## Abstract

**Simple Summary:**

Cytosolic DNA sensors (CDSs), expressed in various types of immune and tumor cells, recognize double-stranded DNA (dsDNA) in the cytoplasm. These molecules are activated after the dsDNA recognition and initiate a cascade of events that culminate in the activation of innate and acquired immunity. CDSs were previously believed to recognize pathogen-derived DNA only. However, they can also respond to cytosolic DNA derived from tumor cells. This suggests that CDSs can be used as small-molecule inhibitor targets in combination with existing immunotherapies to enhance anti-tumor immune responses. This review summarizes current research on the mechanisms underlying CDSs, absent in melanoma 2 (AIM2), cyclic GMP-AMP synthase (cGAS), and stimulator of interferon genes (STING)—downstream signaling effectors of cGAS—in the tumor microenvironment. Furthermore, this review discusses the prospects for future anti-tumor immunotherapy strategies based on these molecules.

**Abstract:**

cGAS and AIM2 are CDSs that are activated in the presence of cytosolic dsDNA and are expressed in various cell types, including immune and tumor cells. The recognition of tumor-derived dsDNA by CDSs in the cytosol of tumor-infiltrating dendritic cells (TIDCs) activates the innate and acquired immunity, thereby enhancing anti-tumor immune responses. STING is the downstream signaling effector of cGAS that induces type I interferon (IFN) signaling. Owing to their ability to activate TIDCs, STING agonists have been intratumorally injected in several clinical trials to enhance the anti-tumor immune response elicited by immune checkpoint antibodies. However, they have shown minimal effect, suggesting the importance of optimizing the dose and route of administration for STING agonists and deciphering other immune pathways that contribute to anti-tumor immune responses. Recent studies have revealed that AIM2 activity induces pro-tumor growth through multiple parallel pathways, including inhibition of STING-type I IFN signaling. Thus, AIM2 could be a potential molecular target for cancer immunotherapies. This review summarizes the current research on the roles of cGAS, STING, and AIM2 in immune cells and tumor cells in the tumor microenvironment and discusses the future prospects of anti-tumor treatment approaches based on these molecules.

## 1. Introduction

Mammalian cells can sense pathogen invasion and induce innate immunity through nucleic acid recognition. When dsDNA enters the cytoplasm of dendritic cells (DCs) and macrophages after a bacterial or viral infection, CDSs are activated to induce an inflammatory response. These include cyclic GMP-AMP synthase (cGAS) and absent in melanoma 2 (AIM2).

Upon activation, cGAS produces cyclic GMP-AMP (cGAMP), which acts as a second messenger. Subsequently, cGAMP binds to and activates STING, a transmembrane protein, in the endoplasmic reticulum. The activated STING molecule moves swiftly from the endoplasmic reticulum to the Golgi body, where it phosphorylates TANK-binding kinase 1 (pTBK1) and recruits interferon regulatory factor 3 (IRF3), a transcription factor. Following its activation through phosphorylation by pTBK1, pIRF3 promotes the expression of type I IFN. This sequence of events is known as the cGAS–STING–type I IFN signaling pathway. Additionally, STING activates nuclear factor-kappa B (NF-κB), a transcription factor that promotes the expression of interleukin (IL)-6, tumor necrosis factor-alpha, and other inflammatory cytokines (Figure 1) [1]. cGAS-deficient mice exhibit higher viral titers and 100% mortality from the *Vaccinia virus* compared with 30% mortality in wild-type mice [2]. In addition, cGAS and STING-deficient mice exhibit increased susceptibility to other DNA viruses, such as murine gammaherpesvirus 68, herpes simplex virus (HSV)1 and HSV2. Collectively, these results suggest that cGAS-STING confers a protective host defense function against DNA viruses [1,2,3,4].

AIM2 senses the DNA of *Listeria*, *Francisella*, and *Mycobacteria tuberculosis*. The AIM2-deficient mice are more susceptible to these intracellular bacteria [5,6,7], which suggests that AIM2 plays an important role in sensing pathogen invasion, inducing innate immunity, and conferring anti-microbial host defense. When AIM2 recognizes and is activated by dsDNA in the cytoplasm, it binds to an apoptosis-associated speck-like protein containing CARD (ASC) and protease caspase-1 (CASP1) to form the AIM2 inflammasome intracellular multiprotein complex. The AIM2 inflammasome cleaves pro-IL-1β, pro-IL-18, and gasdermin D (GSDMD) by activating CASP1. Furthermore, GSDMD comprises one domain on the N-terminal that executes pore-forming activity on the cell membrane and another domain on the C-terminal that suppresses the breakdown of the cell membrane. When GSDMD is cleaved by CASP1, its N-terminal oligomerizes, thereby forming pores in the cell membrane. In addition, IL-1β and IL-18 are released extracellularly after pore formation. When the number of pores exceeds the cell membrane threshold, pyroptosis (a lytic form of cell death) is induced and K^+^ is discharged. This further activates the inflammasome, and the cells release large quantities of IL-1β and IL-18 to the microenvironment (Figure 1) [5,8].

It was previously believed that the cGAS–STING–type I IFN pathway induces the production of type I IFN during DNA virus infections, whereas AIM2 produces IL-1β and IL-18 during intracellular bacterial infections, resulting in an innate immune response that induces pyroptosis. However, cGAS and AIM2 have been found to respond to pathogen-derived DNA, self-DNA, and tumor-derived DNA [1,8]. This indicates that they play important roles in autoimmune diseases and antitumor immunity. Therefore, cGAS, STING, and AIM2 have attracted attention as drug discovery seeds in the cancer immunotherapy field.

## 2. The Role of cGAS and STING in Immune Cells in the Tumor Microenvironment

The advent of immune checkpoint inhibitors (ICIs), such as anti-programmed cell death protein 1 (PD-1) and anti-cytotoxic T-lymphocyte associated protein 4 (CTLA-4) antibodies (Abs), has revolutionized the treatment of patients with advanced melanoma. Five-year overall survival (OS) of patients with melanoma exhibiting distant metastasis (stage IV), treated with anti-PD-1 Ab or anti-PD-1 + anti-CTLA-4 Ab, has been reported at 40% and 50%, respectively [9]. However, responses to this treatment require existing inflammation of the tumor marked by infiltration of CD8^+^ T-cells, a condition known as “hot tumor”. In contrast, “cold tumors” have minimal CD8^+^ T-cell infiltration and exhibit a poor response to ICI therapies [10].

Analyses of hot tumors have revealed that cDC1 (CD24^+^XCR1^+^ DCs in mice, CD141^+^ XCR1^+^ DCs in humans) tumor-infiltrating dendritic cells (TIDCs) recognize tumor-derived DNA in the cytoplasm via cGAS. The cGAS activates the cGAS–STING–type I IFN signaling pathway to promote the priming of T-cells in regional lymph nodes. Furthermore, the type I IFN signaling pathway promotes the intratumoral infiltration of CD8^+^ T-cells via C-X-C motif chemokine ligand 10 (CXCL10), which is induced by type I IFN (Figure 2) [11,12]. Considering the importance of the cGAS–STING–type I IFN signaling pathway in cancer immunity, STING agonists are being developed [13,14]. Indeed, there are several ongoing clinical trials evaluating the efficacy of combination therapy using STING agonists in combination with anti-PD-1 antibodies on solid cancers, including melanoma [13,15]. In addition, recent studies have investigated whether there are existing therapies that can activate the cGAS–STING–type I IFN signaling pathway or increase the efficacy of STING agonists, with the following findings:By using melanoma, colon, and breast cancer mouse models and human cDC1 and cDC2 subsets from the peripheral blood, it was demonstrated that radiation therapy destroys tumor cells, which then release their dsDNA into the tumor microenvironment; cDC1 and cDC2 take up the dsDNA into their own cytoplasm. This action promotes intratumoral CD8^+^ T-cell infiltration through activation of the cGAS–STING–type I IFN–CXCL10 axis [11,16,17]. Similarly, using a breast cancer mouse model, it was shown that chemotherapy using topotecan induces the release of exosomes containing tumor-derived dsDNA from tumor cells [18].By using a colon cancer mouse model and human colorectal cancer specimens, it was demonstrated that tumor cells release their own cGAMP molecules into the tumor’s microenvironment. These molecules are then taken up by cDC1 TIDCs, thereby promoting CD8+ T-cell infiltration into the tumor through activation of the STING type I IFN signaling pathway [19]. In addition, melanoma mouse models and the Cancer Genome Atlas (TCGA) melanoma dataset revealed that the activation of this signaling pathway promotes the intratumoral infiltration of NK cells and enhances the antitumor immune response of NK cells [20].By using a breast cancer mouse model and human cDC1 subsets from the peripheral blood, it was shown that the cDC1 take up the dsDNA bound to high mobility group box 1 (HMGB1), a DNA-binding protein, into their cytoplasm via endocytosis, thereby activating the cGAS–STING type I IFN signaling pathway. In addition, HMGB1-dependent endocytosis is suppressed via the clustering of T-cell immunoglobulin and mucin-containing domain-3 (TIM3) that are expressed on the cellular membrane surface of TIDCs. Galectin-9 regulates TIM3 cell surface clustering and inhibitory function. Therefore, anti-galectin-9 and anti-TIM-3 antibodies promote the HMGB1-dependent endocytosis of dsDNA in vivo, thereby enhancing cGAS–STING–type I IFN signaling pathway activation [17].By using colon cancer and melanoma mouse models, it was demonstrated that STING activates the interferon regulatory factor 3 (IRF3), which accelerates the production of type I IFN. Moreover, STING can activate the classical NF-κB pathway (NF-κB1), which leads to the activation of type I IFN signaling. However, concurrently with the activation of NF-κB1, STING activates the non-classical NF-κB pathway (NF-κB2), which suppresses the production of type I IFN. Therefore, the additional use of an NF-κB2 inhibitor when administering a STING agonist leads to enhanced anti-tumor immune responses [21].
Figure 2Role of type I IFN signaling and interaction between innate and acquired immunity. Type I IFN is produced by DC as well as macrophages in the tumor microenvironment and through the signaling mediated by CDSs within tumor cells. Additionally, type I IFN promotes the intratumoral infiltration of NK cells, which in turn promotes the intratumoral infiltration of cDC1 via XCL1, CCL5, and FMS-like tyrosine kinase 3 ligand (FLT3LG). Furthermore, cDC1 TIDCs induce CXCL9 and CXCL10 secretion in the tumor microenvironment, which in turn promotes the intratumoral infiltration of CD8^+^ T and NK cells.
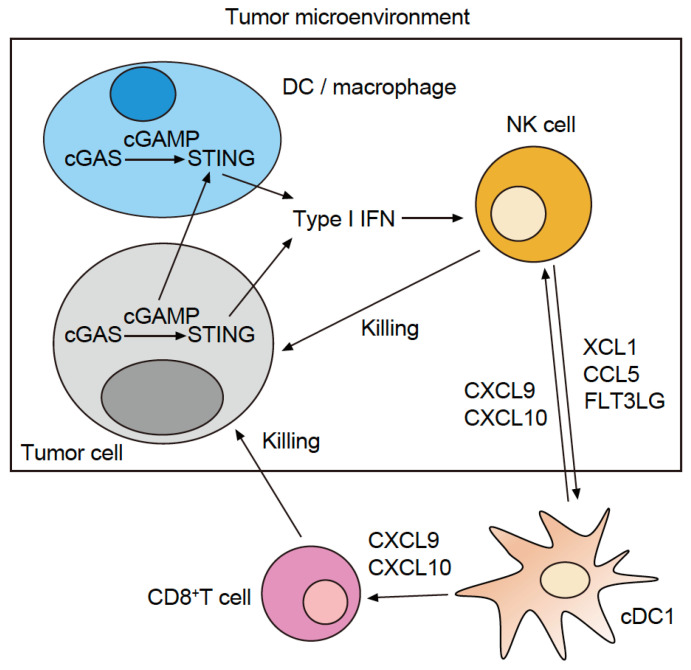



The combination of anti-PD-1 Abs and STING agonists is one of the most tested regimens in clinical trials. Based on the abovementioned results, the addition of (1) anti-galectin-9 Ab, (2) anti-TIM-3 Ab, or (3) NF-κB2 inhibitors to this regimen may be tested in the future. However, in cold tumor cases where only a small number of cDC1 TIDCs have infiltrated the tumor, there is almost no cDC1. Thus, the addition of a treatment strategy that increases cDC1 TIDCs would be optimal for combined therapy with anti-PD-1 Ab and STING agonists. Recent studies of melanoma model mice and the TCGA melanoma dataset have shown that infiltration of cDC1 in the tumor is induced by XCL1, CCL5, or FMS-like tyrosine kinase 3 ligand (FLT3LG), which are produced by the NK cells infiltrated in the tumor (Figure 2) [22,23]. Thus, combination therapy with an intratumoral administration of cDC1 (cultivated in vitro), XCL1, CCL5, or FLT3LG together with the anti-PD-1 antibody + STING agonist would be desirable in the treatment of cold tumors.

## 3. The Role of cGAS and STING in Tumor Cells

Most studies focus on immune cells within tumors. Therefore, data on the function of STING in tumor cells are limited. Studies using mouse and human colon cancer samples, as well as lung cancer model mice and the TCGA lung cancer dataset, found that tumor cells suppress the high-level expression of cGAS and STING through epigenetic regulation or the production of DNase [24,25]. Considering these features, the following new treatment strategy was attempted: Cyclic dinucleotides (CDNs) bonded to STING (which had been altered to prevent degradation by DNase) were introduced into the tumor of a melanoma mouse model. Then, apoptosis was artificially induced via radiation therapy and chemotherapy, causing large numbers of CDNs to be released into the tumor microenvironment. This had a powerful activation effect on the STING type I IFN signaling pathway of TIDCs and demonstrated that CDNs + radiation or chemotherapy could be a novel treatment strategy for activating STING in TIDCs [26].

Chromosomal instability (CIN) is a phenomenon that has been observed in some cancer types, where some or all chromosomes increase or decrease in number. Binding between cGAS and dsDNA is more likely to occur in “CIN-high” cancers than in “CIN-low” cancers. Although STING is activated, the downstream type I IFN signaling pathway is not, resulting in discrepancies, such as a high potential for cancer cell proliferation and metastasis [27]. In CIN-high cancers, IRF3 and NF-κB1 do not activate the type I IFN signaling pathway downstream of STING. Rather, noncanonical NF-κB (NF-κB2) signaling activation suppresses this signaling pathway [27].

This finding suggests that the activation of the type I IFN signaling pathway could be suppressed despite the activation of tumor cell STING signals. Therefore, the treatment should cause the release of cGAMP and CDNs from tumor cells into the tumor’s microenvironment to activate STING in the cDC1 TIDCs that have infiltrated the cancerous growth, thereby enhancing the anti-tumor immune response.

## 4. STING Agonists in Clinical Trials and Next-Generation STING Agonists

The first STING agonist evaluated for efficacy against cancer was DMXAA, which exhibited antitumor activity in preclinical models. However, phase III trials that compared DMXAA in combination with carboplatin and paclitaxel chemotherapies (CP therapy) to only CP therapy for previously untreated non-small cell lung cancer failed to demonstrate an improved response rate, progression-free survival (PFS), or OS [28]. Later studies demonstrated that, although DMXAA can activate mouse STING, it fails to activate any of the five major human STING isoforms [29,30].

Based on previous results of DMXAA, Aduro Biotech has developed MIW815 (ADU-S100), a CDN that agonizes cGAMP, thereby activating human STING isoforms and acting as a human STING agonist. CDNs are susceptible to enzymatic degradation by serum phosphatases and hydrolases [31], leading to low bioavailability; hence, MIW815 was administered intratumorally. In the phase I dose escalation trial of MIW815 monotherapy [14], a total of 47 patients with advanced/metastatic solid tumors or lymphomas were treated with weekly intratumoral injections ranging from 50 to 6400 μg, 3-weeks on and 1-week off. The most frequent treatment-related adverse events (AEs) were on-target expected toxicities, such as pyrexia (17%) and injection site pain (15%). Although clonal T-cell expansion and a dose-dependent increase of IFN-β were observed in peripheral blood, there was no increase in CD8 positivity on treatment in the tumor. One patient (with Merkel cell carcinoma) experienced a partial response, with an overall response rate (ORR) of only 2.1%. Importantly, pharmacokinetic analyses revealed that MIW815 was rapidly absorbed from the injection site into the systemic circulation and had a short half-life of 24 min. Furthermore, a phase Ib dose-escalation trial of MIW815 in combination with anti-PD-1 Ab was conducted [15], in which 106 patients with advanced solid tumors or lymphomas were treated. The combination therapy was well tolerated; however, the antitumor response was minimal, and the ORR was only 10.4%. These results suggested that an improvement in the retention of CDNs in the tumor is necessary to enhance its efficacy. In addition, intratumoral administration of STING agonists showed the impossibility of injecting all metastatic lesions in an advanced cancer patient.

To overcome these challenges, a non-CDN small molecule STING agonist, GSK3745417 (diABZI), that is systemically administered, has been developed [32]. As it showed efficacy for treating a colon cancer mouse model [32], diABZI in combination with anti-PD-1 Ab has undergone clinical trials for advanced solid tumors (NCT03843359) [13]. However, there are two major concerns [33]: Firstly, systemic STING activation may lead to severe toxicity due to the enhanced activation of type I IFN signaling and IL-6 signaling in off-target tissues. To address this problem, non-CDN STING agonists, SR-717 and MSA-2, were developed [34,35]. While diABZI stabilizes STING in its open conformation, SR-717 and MSA-2 can stabilize the closed conformation of STING, similar to endogenous cGAMP. This enables effective doses of SR717 to significantly decrease the serum IFN-β concentration compared with diABZI. MSA-2 also showed limited toxicity in mice due to its effective bioactivity under the acidic conditions of the tumor microenvironment. Secondly, systemic STING agonists may affect specific immune cell subpopulations. Intraperitoneal injection of SR-717 did not enhance the anti-tumor immunity of either anti-PD-1 or anti-PD-L1 therapy in a melanoma mouse model. Immunological analysis revealed that SR-717 increased programmed cell death protein 1 ligand 1 (PD-L1) expression on CD11c^+^CD8^−^ dendritic cells but not on cDC1s in tumor-draining lymph nodes. In contrast, oral MSA-2 significantly decreased the tumor volume of melanoma, colon, and lung cancer mouse models when combined with anti-PD-1 therapy. These results suggest that the molecular properties of STING agonists and administration modes should be optimized to achieve optimal treatment outcomes.

New types of packaged STING-activating CDNs with improved cytosolic delivery and stability are also candidates for future STING agonists (Table 1). Since liposomes and polymeric nanoparticles can overcome biological barriers for enhanced CDN [36], STING-activating nanoparticles (STING-NPs) were developed. Evaluation of their efficacy in a melanoma mouse model and human metastatic melanoma tissue [37] revealed that STING-NPs increase the biological potency of cGAMP and enhance activation of STING-type I IFN signaling in the tumor microenvironment, thereby converting cold tumors to hot tumors. This improved the response rate to anti-PD-1 Ab and overall survival. Furthermore, researchers recently developed CDN prodrugs called lipid nanodiscs (LND-CDNs), a form of nanoparticles linking CDNs to the polyethylene glycol (PEG)-lipid component of their LND [38]. Since the PEG-lipid component and LND are linked with a peptide linker that is cleaved by an intracellular protease highly expressed by tumor cells, LDN-CDNs increased CDN release in the tumor microenvironment of colon and breast cancer model mice and decreased off-target toxicity. In addition, LND-CDN uptake by cancer cells correlated with robust T-cell activation, and a single dose of LND-CDNs led to the rejection of murine colon and breast cancer models. Collectively, LND-CDNs are a promising vehicle for delivery of STING-activating CDNs throughout solid tumors, which can be used as an adjuvant for enhanced ICI therapies.

## 5. The Role of AIM2 in Tumor-Infiltrating Dendritic Cells

A phase I anti-PD-1 antibody + STING agonist trial has reported a slight increase in the antitumor immune response [15], which suggests that in addition to STING-dependent type I IFN production, other immune pathways contribute to immune infiltration into tumors. This demonstrates the importance of conceiving a therapeutic strategy that increases the efficacy of cancer immunotherapy. Although much attention has been paid to cGAS and STING in TIDCs, there have been few studies analyzing the function of TIDC-resident AIM2, which is also a cytosolic dsDNA sensor. Through our analysis of a mouse model of cold melanoma and melanoma specimens from patients, we revealed that AIM2 exerts an immunosuppressive effect within the melanoma microenvironment in mice and correlates with tumor progression in human melanoma patients [39]. Intravenous administration of AIM2-deficient dendritic cells (DCs) using a vaccination strategy results in the homing of the DCs to the tumor and enhanced efficacy of adoptive cell therapy (ACT) as well as anti-PD-1 immunotherapy, thereby achieving therapeutic responses in cold melanomas. This effect depends on tumor-derived DNA activating STING-dependent type I IFN secretion and the subsequent production of CXCL10 to recruit CD8^+^ T-cells. In addition, loss of AIM2-dependent IL-1β and IL-18 processing further enhances the treatment response by limiting the recruitment of T regulatory cells (Tregs). Hence, AIM2-deficient DC vaccination not only enhances immune responses to the tumor by activating STING but also suppresses IL-1β and IL-18 production, resulting in synergistic therapeutic responses (Figure 3).

Unlike that in mice, cGAS in human monocytes recognizes cytosolic dsDNA, activating the cGAS–STING–type I IFN signaling pathway and the NLRP3 inflammasome, thereby resulting in the release of IL-1β and IL-18 [40]. Thus, it was surmised that AIM2 inflammasomes in human monocytes do not produce IL-1β or IL-18 in response to cytosolic dsDNA. However, subsequent studies found that the AIM2 molecules in human monocyte-derived DCs [39] and macrophages [41] recognized cytosolic dsDNA, which led to IL-1β and IL-18 production. Moreover, activation of the cGAS–STING–type I IFN signaling pathway was increased when AIM2 was inhibited. These findings suggest that anti-PD-1 antibodies combined with an Aim2-suppressed DC vaccine or Aim2 siRNA could be a new treatment strategy for patients with cold melanoma (Figure 3) [39].

Similar to AIM2-deficient BMDCs, ASC, CASP1, and GSDMD-deficient BMDCs accelerate the activation of the cGAS–STING–type I IFN signaling pathway in response to dsDNA to varying degrees [42]. Studies using GSDMD-deficient BMDCs and bone marrow-derived macrophages have shown that the release of K^+^ from the cell membrane pores formed by GSDMD suppresses the activation of cGAS, thereby regulating the accelerated activation of the cGAS–STING–type I IFN signaling pathway [42]. This suggests that the AIM2 inflammasome and GSDMD regulate the expression of the cGAS–STING–type I IFN signaling pathway. Therefore, therapy targeting ASC, CASP1, and GSDMD in TIDCs may also serve to convert cold melanomas to hot melanomas.

## 6. The Role of AIM2 in Immune Cells of the Tumor Microenvironment, Other Than in Tumor-Infiltrating Dendritic Cells

More research has been conducted in recent years on the CDS found in tumor-associated macrophages (TAMs). Tumor cells that are bonded to antibodies (molecularly targeted drugs), such as the anti-human epidermal growth factor receptor 2 (anti-HER2) and anti-CD20 antibodies, are taken up into TAMs as a result of antibody-dependent cellular phagocytosis (ADCP). After ADCP, TAMs inhibit NK cell-mediated antibody-dependent cellular cytotoxicity (ADCC) and T-cell-mediated cytotoxicity in a breast cancer and lymphoma mouse model. Moreover, AIM2 is recruited to the phagosomes through Fcγ receptor signaling and activated by sensing the phagocytosed tumor DNAs through the disrupted phagosome membrane. This subsequently activates IL-1β signaling, which induces the expression of the immune checkpoint molecules PD-L1 and indoleamine 2,3-dioxygenase (IDO) on the TAM cell surface. These results revealed the role of ADCP of TAMs in cancer immunosuppression and suggested that simultaneous therapeutic antibodies and AIM2 inhibition of TAMs provide synergistic effects in cancer treatment [43].

Although AIM2 has long been investigated for its function in bone marrow cells, such as DCs and macrophages, it is more strongly expressed in Tregs. Therefore, there has been a new focus on its function in this class of T-cells. In experimental autoimmune encephalomyelitis and chronic inflammation in enteritis, the expression of AIM2 by Tregs maintains the expression of the transcription factor forkhead box P3 (FOXP3). In other words, AIM2 maintains Treg homeostasis in cases of inflammation. As AIM2 binds with the receptor for activated C kinase 1 (RACK1) and protein phosphatase 2A (PP2A), it suppresses activation of the protein kinase B–mammalian target of the rapamycin (AKT–mTOR) pathway. This signaling pathway promotes the expression of the cancer gene *c-myc* and, conversely, suppresses the expression of FOXP3. Therefore, when AIM2 is removed from the Tregs at sites of chronic inflammation, the AKT–mTOR pathway is activated, *c-myc* expression is increased, and FOXP3 expression is decreased. This reduces the number of bona fide Tregs. Moreover, multiple models of chronic inflammation have shown that increasing the amount of IFN-γ produced by Tregs and decreasing the immunosuppressive action exacerbates inflammation. These findings suggest that, in addition to its suppression in TIDCs, the suppression of AIM2 in TAMs and Tregs also increases antitumor immune response [44].

## 7. The Role of AIM2 in Tumor Cells

AIM2 is hardly expressed in melanomas and was initially discovered as a tumor suppressor gene [45]. Most melanoma cells silence the expression of one or more inflammasome components and do not produce IL-1β by themselves; rather, they induce IL-1β production from TAMs by releasing endogenous danger signals [46]. Hence, there is a low probability that drugs targeting AIM2 will act on melanoma cells.

The functions of AIM2 have been investigated in many cancer types, including melanoma. For instance, AIM2 has long been known as a marker for poor prognosis in colorectal cancer. This CDS suppresses the phosphoinositide 3-kinase (PI3K)–AKT signaling pathway in colorectal cancer cells. In patients with colorectal cancer with low AIM2 expression, the PI3K–AKT signaling pathway is upregulated, which increases the proliferative and survival abilities of the cancer cells, making them highly malignant. In addition, research using a colon cancer mouse model revealed that low AIM2 expression induced an imbalance of the intestinal flora (dysbiosis), thereby providing a suitable environment for the proliferation of colorectal cancer cells [47]. Similarly, breast cancer, hepatocellular carcinoma, and renal cell carcinoma exhibit low levels of AIM2 expression, promoting cancer cell proliferation and metastasis [2]. In contrast, cancer cell proliferation and metastasis are suppressed when AIM2 expression is low in patients with cutaneous squamous cell carcinoma and NSCLC [2]. These findings indicate that the administration of an AIM2 inhibitor may increase the malignancy of the tumor. Therefore, the use of drugs that inhibit this CDS increases tumor malignancy in certain cancer types and should be limited to the types of cancer in which their administration suppresses the proliferation and metastasis of the tumor cells.

## 8. Conclusions

STING agonists that activate cDC1 TIDCs have been investigated as adjuvants for increasing the antitumor immune response elicited by anti-PD-1 immunotherapy. However, the results of combination therapy with anti-PD-1 antibodies + intratumoral STING agonist administration have not been promising. Therefore, STING agonists that can be administered systemically and stabilize the close confirmation of STING to decrease the toxicity may be an effective approach for targeting the STING-type I IFN signaling pathway. Combination therapy consisting of the anti-PD-1 Ab+ systemic STING agonist should be investigated in future clinical trials since it has the potential to transform the therapeutic landscape once optimized.

In addition, there remains a need to analyze other immune pathways that contribute to the immune infiltration into tumors and thus help to determine their “hot” or “cold” states. Researchers have begun testing antibodies against TIM-3, which is an immune checkpoint receptor that suppresses the efficacy of STING agonists and regulates CD8^+^ T-cells. As there is no drug that selectively inhibits AIM2, it would be impossible for a treatment that targets this molecule to be put to clinical use in the near future. In addition to being used as a STING agonist, anti-AIM2 therapy would activate TIDCs and act as an IL-1β and IL-18 inhibitor, thus acting on more pathways than the STING agonist does. Therefore, as a candidate for molecularly targeted drug therapy, AIM2 could be more effective at reinforcing antitumor immunotherapies. Researchers have recently reported the engineering of a nanomolecular STING agonist vaccine that has been modified to be taken up by cDC1 [48]. An AIM2 siRNA nanovaccine developed using this same technology could enhance the immunotherapeutic antitumor effect of the anti-PD-1 Ab. Moreover, the suppression of AIM2 in TIDCs, macrophages, and Tregs is expected to greatly enhance antitumor immunotherapies. Thus, an AIM2 siRNA nanovaccine that acts selectively on CD45^+^ cells could be a novel cancer immunotherapy.

## Figures and Tables

**Figure 1 cancers-15-02114-f001:**
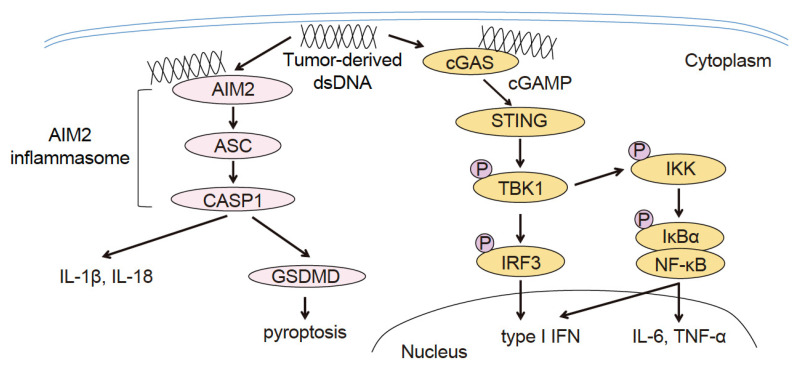
Innate immune signaling pathway that recognizes cytosolic DNA in tumor-infiltrating dendritic cells (TIDCs).

**Figure 3 cancers-15-02114-f003:**
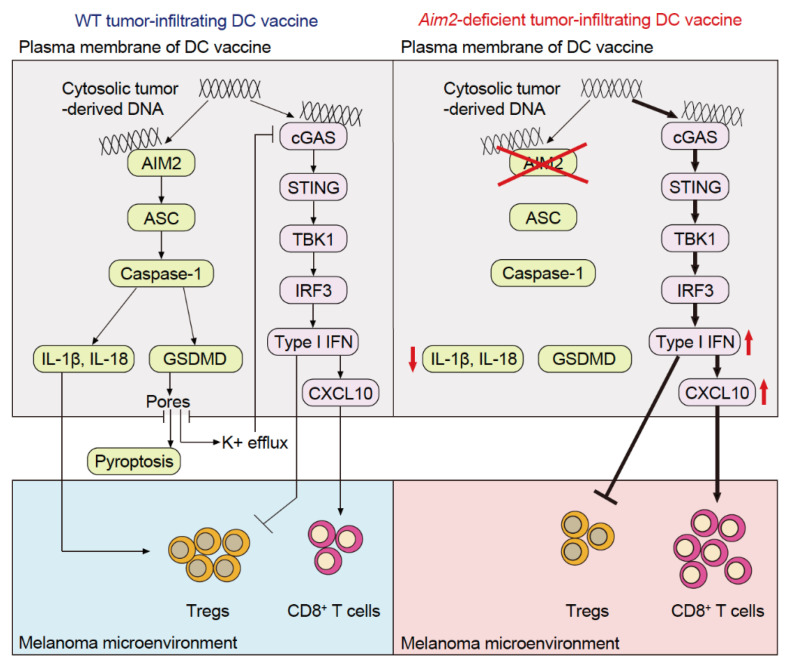
Tumor immunosuppressive mechanism of AIM2 in TIDCs. Upon recognizing the tumor-derived DNA taken up into the TIDC cytoplasm, AIM2 induces the production of IL-1β and IL-18, which in turn promote the intratumoral infiltration of regulatory T-cells (Tregs). Additionally, AIM2 suppresses STING type I IFN signaling, a pathway that promotes intratumoral infiltration of CD8^+^ T-cells and suppresses that of Tregs.

**Table 1 cancers-15-02114-t001:** Clinical trials of Sting agonist and future STING agonists.

Drug	Route	Combination Therapy	Indication	NCT Identifier	Study Phase	Status	Reference
DMXAA	IV	carboplatin ± paclitaxel	Advanced NSCLC	NCT00662597	III	Terminated	[27]
MIW815 (ADU-5100)	IT	± anti-CTLA-4 Ab	Advanced solid tumors or lymphoma	NCT02675439	1	Terminated	[14]
	IT	+ anti-PD-1 Ab	Advanced solid tumors or lymphoma	NCT03172936	I	Terminated	[15]
GSK3745417 (diABZI)	IV	+ anti-PD-1 Ab	Advanced solid tumors	NCT03843359	1/11	Ongoing	[13]
SR-717	IP	+ anti-PD-1 Ab + anti-PD-L1 Ab	melanoma (mouse)		preclinical		[33]
MSA-2	PO	+ anti-PD-1 Ab	melanoma, colors cancer, lung cancer (all mouse)		preclinical		[34]
STING-NPs	IV	+ anti-PD-1 Ab	melanoma (mouse, human)		preclinical		[35]
LND-CDNs	IV	+ anti-PD-1 Ab	colors cancer, breast cancer (both mouse)		preclinical		[36]

Abbreviations: IV, intravenous; IT, intratumoral; IP, intraperitoneal; PO, per os (oral); AB, antibody; NSCLC, non-small cell lung cancer.

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
