# Peer review of "Immune Regulation by Cytosolic DNA Sensors in the Tumor Microenvironment"

_cancers, 2023, doi:10.3390/cancers15072114_

Round 1

Reviewer 1 Report

The review article by K. Fukuda entitled “Immune regulation by cytosolic DNA sensors in the tumor microenvironment” is well written with comprehensive yet concise summary of recent advances on this topic, which include key studies and findings in recent years. This timely review is highly relevant to the cytosolic DNA sensing, the STING pathway and cancer immunotherapy, which will be of high interest to a large group of readers in this research area. There are a few points as detailed below, which should strengthen the review once addressed or clarified.

Figure 1:

It would be helpful to add the NF-kB pathway in the diagram, which is activated by pTBK1 and can promote type I IFN production together with IRF3, as well as the production of IL-6 and TNF-a.

There is no Figure 2.

Figure 3: should change to Figure 2

1) It is not cGAS, but cGAMP, in tumor cells that is transmitted to DC/macrophage; therefore, the line pointing cGAS from tumor cell (gray) to DC/macrophage (blue) should be removed.

2) It showed that NK cells produce XCL1 in the figure, but in the figure legend it is CXCL1. Should change to XCL1. Additionally, should add NK in the last sentence of the figure legend (…infiltration of CD8+ T and NK cells), which is indicated by a line next to CXLC9/CXCL10 from cDC1 to NK cell in the figure.

Figure 4: should be Figure 3

It is not clear from the figure how AIM2 inhibits STING-type I IFN signaling as stated in the figure legend, by competing with cGAS?

Line 84: Markers for murine cDC1 include XCR1 and CD24. CD103 is a marker for migratory cDC1; in another word, CD103+ cDC1 is a subgroup of all cDC1.

Line 85: Is there a difference between dsDNA derived from tumor vs normal cells?  

Line 98: Is cDC1 the only DC subgroup that can uptake tumor-derived dsDNA? How about cDC2?

Lines 104 and 168: Based on the citation, it is not cGAS, but cGAMP, that is released by tumor cells into the tumor microenvironment.

Lines 132 and 134: Should it be XCL1, not CXCL1? Should also add citation for NK cells producing XCL1 and CCL5.

Line 150: can add reference Kitajima S et al., Cancer Discovery, 2019, 9:34-45 after reference 18.

Lines 250-254: The logic of these statements is confusing, not clear.

Line 258: add reference [31] after “…within the melanoma”.

Line 304: add “of” between ADCP and TAMs- “the role of ADCP of TAMs”.

Line 343: should add “…increase the malignancy of the tumor in certain cancer types”.

Line 362: Why anti-AIM2 therapy would suppress TIDCs? Shouldn’t it be the opposite?

References: There are two sets of numbers after reference 3, and the one on the left is wrong.

Reviewer 2 Report

The review is very well written, rich in particular biological mechanisms that the authors highlight and

connect in a very interesting way. However, it is necessary to avoid redundancies and to be more precise at

certain points in the text.

Introduction

The introduction is well written but very summarizing the knowledge about the main markers implied in

studied mechanisms, please deepen the role of these proteins.

Line 96-97: I suggest, if the authors agree, to include a table or a scheme that summarize the different

approaches described as a list

Please summarize in a table or report in the text the registration number of clinical trials, to which the

authors refer

In the text is important report and highlight when an evidence is obtained in vivo, in vitro or in ex vivo

model, it is important to allow to understand the real effectiveness, the importance and the applicability of the presented mechanisms
